# Synthesis, Crystal Structure, Thermal Analysis, and DFT Calculations of Molecular Copper(II) Chloride Complexes with Bitopic Ligand 1,1,2,2-tetrakis(pyrazol-1-yl)ethane

**Elizaveta Lider** [1,2], **Taisiya Sukhikh** [1,2], **Anton Smolentsev** [1], **Evgeny Semitut** [1],
**Evgeny Filatov** [1,2] **and Andrei Potapov** [1,3,*]

[1] Nikolaev Institute of Inorganic Chemistry, Siberian Branch of the Russian Academy of Sciences, 3 Lavrentiev Ave., 630090 Novosibirsk, Russia; lisalider@ngs.ru (E.L.); sukhikh@niic.nsc.ru (T.S.); smolentsev@ngs.ru (A.S.); semitut@ngs.ru (E.S.); decan@niic.nsc.ru (E.F.)

[2] Department of Natural Sciences, Novosibirsk State University, 2 Pirogova Str., 630090 Novosibirsk, Russia

[3] Kizhner Research Center, National Research Tomsk Polytechnic University, 30 Lenin Ave., 634050 Tomsk, Russia

* Correspondence: potapov@niic.nsc.ru

**Abstract:** Two binuclear coordination compounds of Cu(II) chloride with the bitopic ligand 1,1,2,2-tetrakis(pyrazol-1-yl)ethane ($Pz^4$) of the composition $[Cu_2(\mu_2\text{-}Pz^4)(DMSO)_2Cl_4]\cdot 4H_2O$ and $[Cu_2(\mu_2\text{-}Pz^4)(DMSO)_2Cl_4]\cdot 2DMSO$ were prepared and characterized by elemental analysis, IR spectroscopy, thermogravimetric analysis, single-crystal X-ray diffraction, and powder diffraction analysis. It was shown that in contrast to silver(I) and copper(II) nitrates, copper(II) chloride forms discrete complexes instead of coordination polymers. The supramolecular structure of the complex $[Cu_2(\mu_2\text{-}Pz^4)(DMSO)_2Cl_4]\cdot 4H_2O$ with lattice water molecules is formed by OH···Cl and OH···O hydrogen bonds. Density functional theory (DFT) calculations of vibrational frequencies of the ligand and its copper(II) complex allowed for assigning IR bands to specific vibrations.

**Keywords:** 1,1,2,2-tetrakis(pyrazol-1-yl)ethane; copper(II) chloride; bitopic ligand; thermogravimetric analysis; binuclear complexes; crystal structure

## 1. Introduction

Multitopic ligands are a class of ligands that nowadays have attracted an increasing amount of interest in many applications of coordination chemistry. These ligands contain two or more separated metal-binding sites that allow them to form a wide variety of structures [1–3]. The design of a linker structure between the binding sites permits tuning the properties of the materials based on coordination compounds with multitopic ligands [4–7]. Applications of functional materials containing such ligands include gas storage [8–10], membrane separations [11], electrochemical devices [12], sensing materials [13,14], drug delivery systems [15,16], enantiomer separations [17], and catalysis in fine organic synthesis [18,19].

Bitopic bis- and tris(pyrazol-1yl)methanes have served as ligands for the preparation of various silver coordination polymers of different topologies [20–28]. In our recent work, we synthesized silver coordination polymers containing the bitopic ligand 1,1,2,2-tetra(pyrazol-1-yl)ethane ($Pz^4$) and showed that silver-based compounds are more likely to form polymeric rather than discrete structures [29]. For the case of $Cu^{2+}$ ions, we demonstrated that both molecular copper(II) complexes and coordination polymers can be prepared [30,31]. Catalytic activity of the binuclear palladium(II) complex of $Pz^4$ [32]

as well as antibacterial and antifungal activity of the ligand itself [33] were reported recently. In this work, we focused on the synthesis and crystal structure peculiarities of Cu(II) coordination compounds containing the $Pz^4$ ligand. Herein, we report the first X-ray crystal structures of the pure ligand and two new copper(II) coordination compounds and discuss their peculiarities.

## 2. Materials and Methods

### 2.1. Instrumental Characterization Methods

Elemental analysis (C, H, N, and S) was carried out on Euro EA 3000 analyzer (Eurovector SPA, Redavalle, Italy) following the standard procedures.

IR spectra of the complexes as mineral and fluorinated oil mulls or polyethylene pellets were recorded on Scimitar FTS 2000 (Digilab LLC, Canton, MA, USA) and Vertex 80 (Bruker Corporation, Billerica, MA, USA) spectrometers in the range of 4000–100 $cm^{-1}$.

X-ray powder diffraction (XRD) patterns were recorded at room temperature on a DRON-RM4 diffractometer (Burevestnik, Saint Petersburg, Russia) using $CuK_\alpha$ irradiation and graphite monochromator $d_{001} = 3345$ Å. The scanning range was 5°–60° (2θ) for complex salts and 5°–90° (2θ) for products of thermolysis. The experimental diffraction data were processed using the PowderCell v.2.4 program [34], which allowed a calculation of the quantitative phase composition and the lattice parameters. Data from the PDF database [35] were used as reference.

Thermogravimetric analysis of coordination compounds was carried out in helium atmosphere on a NETZSCH thermobalance TG 209 F1 Iris (Erich NETZSCH GmbH & Co. Holding KG, Selb, Germany) in open $Al_2O_3$ crucibles (loads of 5–10 mg, heating rate of 10 K·$min^{-1}$).

### 2.2. X-ray Structure Determination

The single crystals of compounds $Pz^4$, **1**, and **2** were selected directly from the mother liquors and mounted on glass fibers using epoxy resin. Single-crystal X-ray diffraction data were collected on a Bruker-Nonius X8 APEX CCD diffractometer (graphite monochromatized Mo Kα radiation, $\lambda = 0.71073$ Å, $\varphi$ and $\omega$ scans of narrow frames (Bruker Corporation, Billerica, MA, USA), equipped with a 4K CCD area detector (Table 1). Absorption corrections were applied using the SADABS program [36]. The crystal structures were solved by direct methods and refined by full-matrix least-squares techniques with the use of the SHELXTL package [37] and Olex2 GUI [38]. Atomic thermal displacement parameters for nonhydrogen atoms were refined anisotropically. The positions of H atoms were calculated corresponding to their geometrical conditions and refined using the riding model.

### 2.3. Computational Chemistry

Experimental X-Ray structures of the ligand $Pz^4$ and complex **1** (without uncoordinated dimethyl sulfoxide (DMSO) molecules) were used as starting points for density functional theory (DFT) geometry optimizations. Singlet- (for $Pz^4$) or triplet-state (for complex **1**) gas-phase geometry optimizations were carried out at the DFT level of theory employing the three-parameter hybrid B3LYP functional [39–42] and 6-31+G(d) basis set [43] in the Gaussian 09 package [44]. Frequency calculations were performed for both molecules in order to ensure the lack of imaginary vibration frequencies, which indicates that the optimized structures correspond to minima on the potential energy surfaces. Cartesian atomic coordinates for all of the optimized structures are provided in Supplementary Tables S1 and S2.

Hirshfeld promolecular surfaces mapped over $d_{norm}$ plots of the complexes were built using the Crystal Explorer (version 17.5) program [45].

## 2.4. Synthesis of Compounds

Analysis-grade copper(II) chloride dihydrate was used for the synthesis of the complexes. $Pz^4$ was prepared as described previously [46]. Solvents were of reagent-grade purity and were used as received.

### 2.4.1. Synthesis of $[Cu_2(\mu_2-Pz^4)(DMSO)_2Cl_4]\cdot4H_2O$ (**1**)

A hot suspension containing $Pz^4$ (0.3 mmol, 0.09 g) in 5 ml of DMSO was added with stirring to an ethanol solution (7 ml) of $CuCl_2\cdot2H_2O$ (1.5 mmol, 0.25 g). Green crystals of **1** suitable for X-ray diffraction analysis were obtained by slow crystallization over the course of 24 h. The light-green precipitate formed from the solution was filtered, washed with ethanol, and then dried in the air. The yield of the product was 0.13 g (55%). Elemental analysis: found, %: C 27.6; H 4.2; N 14.4; S 8.0; for $C_{18}H_{34}Cl_4Cu_2N_8O_6S_2$ calculated, %: C 27.3; H 4.3; N 14.2; S 8.1. IR bands, cm$^{-1}$: 3513, 3424 ($\nu$OH ($H_2O$)), 3125, 3101, 2987 ($\nu$CH (Pz)), 1630 ($\delta$OH ($H_2O$)), 1513 ($\nu$CH (Pz)), 467 ($\nu$(Cu–O)), 295 ($\nu$(Cu–N)), 254 ($\nu$(Cu–Cl)).

**Table 1.** Crystallographic data of the compounds 1,1,2,2-tetra(pyrazol-1-yl)ethane ($Pz^4$), **1**, and **2**.

| Identification Code | $Pz^4$ | 1 | 2 |
|---|---|---|---|
| Empirical formula | $C_{14}H_{14}N_8$ | $C_{18}H_{34}Cl_4Cu_2N_8O_6S_2$ | $C_{22}H_{38}Cl_4Cu_2N_8O_4S_4$ |
| Formula weight | 294.33 | 791.53 | 875.72 |
| Temperature/K | 150(2) | 296(2) | 150(2) |
| Crystalsystem | monoclinic | triclinic | monoclinic |
| Spacegroup | $C2/c$ | $P-1$ | $P2_1/c$ |
| a/Å | 15.0087(9) | 7.959(3) | 8.7218(4) |
| b/Å | 5.4385(3) | 8.885(3) | 18.2205(7) |
| c/Å | 17.3736(11) | 12.269(5) | 11.9231(5) |
| $\alpha/°$ | 90 | 73.239(16) | 90 |
| $\beta/°$ | 92.435(2) | 72.615(16) | 107.0408(16) |
| $\gamma/°$ | 90 | 83.178(18) | 90 |
| Volume/Å$^3$ | 1416.84(15) | 792.3(5) | 1811.58(13) |
| Z | 4 | 1 | 2 |
| $\rho_{calc}$g/cm$^3$ | 1.380 | 1.659 | 1.605 |
| $\mu$/mm$^{-1}$ | 0.092 | 1.857 | 1.740 |
| F(000) | 616.0 | 404.0 | 896.0 |
| Crystalsize/mm$^3$ | $0.35 \times 0.1 \times 0.09$ | $0.25 \times 0.15 \times 0.1$ | $0.2 \times 0.12 \times 0.1$ |
| 2Θ range for data collection/° | 4.694–55.148 | 4.792–55.588 | 4.214–52.818 |
| Index ranges | $-19 \leq h \leq 14$, $-7 \leq k \leq 4$, $-20 \leq l \leq 22$ | $-9 \leq h \leq 10$, $-11 \leq k \leq 11$, $-14 \leq l \leq 15$ | $-10 \leq h \leq 10$, $-22 \leq k \leq 22$, $-14 \leq l \leq 14$ |
| Reflections collected | 3290 | 6095 | 24618 |
| Independent reflections | 1628 [$R_{int}$ = 0.0167, $R_{sigma}$ = 0.0257] | 3598 [$R_{int}$ = 0.0391, $R_{sigma}$ = 0.0579] | 3715 [$R_{int}$ = 0.0328, $R_{sigma}$ = 0.0217] |
| Restraints/parameters | 0/100 | 0/189 | 0/203 |
| Goodness-of-fit on F$^2$ | 1.053 | 1.044 | 1.035 |
| Final R indexes [I ≥ 2σ (I)] | $R_1$ = 0.0386, w$R_2$ = 0.0964 | $R_1$ = 0.0424, w$R_2$ = 0.1092 | $R_1$ = 0.0252, w$R_2$ = 0.0556 |
| Final R indexes [all data] | $R_1$ = 0.0494, w$R_2$ = 0.1029 | $R_1$ = 0.0584, w$R_2$ = 0.1153 | $R_1$ = 0.0313, w$R_2$ = 0.0580 |
| Largest diff. peak/hole / e Å$^{-3}$ | 0.31/−0.23 | 0.70/−0.47 | 0.74/−0.30 |

### 2.4.2. Synthesis of $[Cu_2(\mu_2-Pz^4)(DMSO)_2Cl_4]\cdot2DMSO$ (**2**).

A suspension of 26.0 mg $Pz^4$ (0.09 mmol) in 1 ml of DMSO was added to 34.0 mg of $CuCl_2\cdot2H_2O$ (0.2 mmol) in a glass vial. The mixture was stirred for 10 min at room temperature. During the stirring process, a light-green precipitate formed. The vial with the formed powder was placed into an oven at 95 °C. After 18 h of heating, the vial was allowed to cool to room temperature. After one day, the crystals initially formed on the bottom of the vial were filtered and washed once with 1 ml of DMSO and then air-dried for a few days. The yield of the product was 39.9 mg (52%). Elemental analysis: found, %: C 29.6; H 4.0; N 12.9; S 14.3; for $C_{22}H_{38}Cl_4Cu_2N_8O_4S_4$ calculated, %: C 30.2; H 4.4; N 12.8; S 14.6. IR bands, cm$^{-1}$: 3141, 3094, 3000 ($\nu$CH (Pz)), 1513 ($\nu$CH (Pz)), 481 ($\nu$(Cu–O)), 294 ($\nu$(Cu–N)), 260 ($\nu$(Cu–Cl)).

## 3. Results and Discussion

### 3.1. Synthesis of the Complexes

The two copper(II) chloride complexes with ligand $Pz^4$ were synthesized by the reaction of the ethanol solution of the $CuCl_2 \cdot 2H_2O$ (for **1**) or solid copper salt (for **2**) with the DMSO suspension of 1,1,2,2-tetrakis(pyrazol-1-yl)ethane according to Scheme 1.

**Scheme 1.** Synthesis route of copper chloride complexes **1** and **2**. i: DMSO-EtOH, 25 °C (for complex **1**); ii: DMSO, 95 °C (for complex **2**).

The synthesis of complex **1** was carried out with stirring at room temperature followed by slow crystallization from solution over the course of 24 h. Complex **1** was synthesized at the ratio $Cu^{2+}:Pz^4 = 5:1$. Complex **2** was obtained after 18 h of heating at 95 °C of the reaction mixture using solid copper salt at the $Cu^{2+}:Pz^4$ ratio of 2.2:1. Although complexes **1** and **2** differed only in outer-sphere solvate molecules, they demonstrated different crystal packing and different thermal behavior.

### 3.2. Crystal Structures of the Complexes

The compounds $[Cu_2(Pz^4)(DMSO)_2Cl_4] \cdot 4H_2O$ (**1**) and $[Cu_2(Pz^4)(DMSO)_2Cl_4] \cdot 2DMSO$ (**2**) include water and DMSO solvate molecules, respectively, and are not isostructural. Both complexes are centrosymmetric binuclear (Figure 1) and reveal very similar molecular geometries. Cu atoms coordinate two $Cl^-$ ligands, one O atom of DMSO ligand, and two N atoms of $Pz^4$. Addison's $\tau_5$ criterion [47] was used to determine the geometry of coordination polyhedra of copper(II) ions in complexes **1** and **2** (Table 2). As suggested by the $\tau_5$ criterion, the environment of the central atom in complex **1** is a slightly distorted trigonal bipyramid with O and N atoms located in axial positions, while in complex **2**, the arrangement is more distorted toward square pyramidal. In previously reported structures of two binuclear copper(II) nitrate polymorphs $[Cu_2(Pz^4)(H_2O)_2(NO_3)_4]$ [30,31], the $\tau_5$ criterion had values in the range of 0.01–0.19, indicative of a square planar geometry of coordination sphere comprising two monodentate nitrate ions, two nitrogen atoms of the ligand, and one water molecule. Thus, the $Pz^4$ ligand shields the central atom in a chelate mode, leaving enough space for three more ligands (chloride/nitrate and solvent molecules). However, it can provide space even for four ligands, as shown by the example of polymeric nitrate complexes $[Cu(Pz^4)(NO_3)_2]_n$ and $[\{Cu(Pz^4)(H_2O)(NO_3)_2\}_2]_n$ [31], which reveal a highly distorted (4+2) octahedral environment. Copper–donor-atom bond lengths and bond angles are very similar in both complexes **1** and **2**, with the exception of the Cu–Cl distance being larger by 0.10 Å for complex **2** in respect to **1**. The metallocycles in **1**, **2**, and known copper(II) complexes with $Pz^4$ have similar geometries, implying their inflexibility (Supplementary Materials Figure S1). The relative positions of one pyrazole unit and DMSO ligand are slightly different for **1** and **2** (Supplementary Materials Figure S2), which is likely due to intermolecular interactions with the solvent molecules and/or packing effects. Analysis of normalized contact distance ($d_{norm}$) mapping on the promolecular Hirshfeld surface of the $Pz^4$ ligand revealed intramolecular C–H⋯Cl contacts between the ethane unit of $Pz^4$ and chloride in both compounds **1** and **2** (Figure 2), being noticeably shorter for the latter. The same type of C–H⋯O contacts with nitrates in place of chlorides has been observed in known complexes $[Cu(Pz^4)(NO_3)_2]_n$ and $[\{Cu(Pz^4)(H_2O)(NO_3)_2\}_2]_n$ [31]. In other words, $Cl^-$ and $NO_3^-$ are inclined to C–H of the ethane unit. However, two polymorphs $[Cu_2(Pz^4)(H_2O)_2(NO_3)_4]$ [30,31] show different arrangement of nitrates, which lie on both sides of the corresponding C–H line. In this way, they are inclined to C–H of the pyrazole unit. Considering the

large distance between C–H and the donor atoms as well as the very low polarity of the C–H bonds, these types of contacts are likely to have a steric nature.

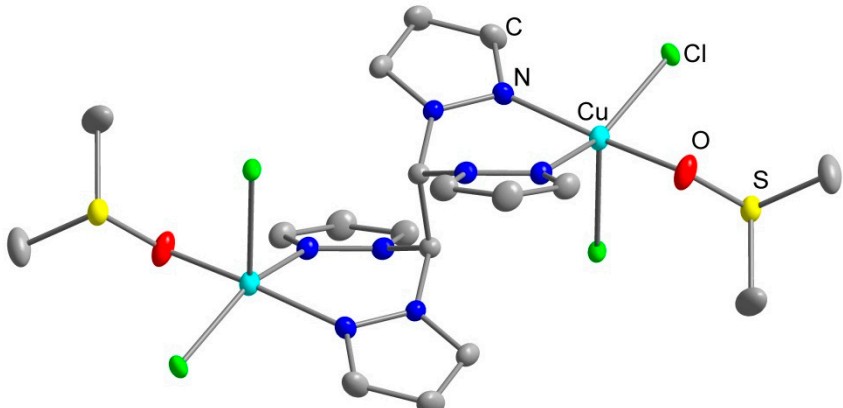

**Figure 1.** Atomic displacement ellipsoid plot of the complexes $[Cu_2(Pz^4)(DMSO)_2Cl_4]\cdot4H_2O$ (**1**) and $[Cu_2(Pz^4)(DMSO)_2Cl_4]\cdot2DMSO$ (**2**) on the example of the latter. Uncoordinated solvent molecules and hydrogen atoms are omitted for clarity.

**Table 2.** Addison's $\tau_5$ criterion for coordination centers in the complexes in copper(II) complexes with $Pz^4$ ligand.

| Parameter | α | β | $\tau_5 = (\beta - \alpha)/60$ | References |
|---|---|---|---|---|
| $[Cu_2(Pz^4)(DMSO)_2Cl_4]\cdot4H_2O$ (**1**) | 129.0 | 171.7 | 0.71 | This paper |
| $[Cu_2(Pz^4)(DMSO)_2Cl_4]\cdot2DMSO$ (**2**) | 153.1 | 168.7 | 0.26 | This paper |
| $[Cu_2(Pz^4)(H_2O)_2(NO_3)_4]$ | 176.0, 178.3 | 168.1, 172.2 | 0.06, 0.17 | [30] |
| $[Cu_2(Pz^4)(H_2O)_2(NO_3)_4]$ | 179.3, 176.1, 177.2, 179.5 | 170.2, 175.4, 173.1, 168.2 | 0.15, 0.01, 0.07, 0.19 | [31] |

In complex **1**, both crystallographically independent Cl atoms form weak intermolecular hydrogen bonds with two water molecules (Figure 3) (O···Cl distances are 3.21 and 3.35 Å). Hydrogen bonds between water molecules were also observed (O···O distances are 2.78 and 2.80 Å), revealing a layered supramolecular structure. Analysis of the Hirshfeld surface revealed intermolecular contacts C–H···D (D = Cl, O) between molecules of the complexes and solvates in **1** and **2** (Supplementary Materials Figure S3) as well as in known copper(II) complexes [30,31], but they are likely to have a steric nature due to the low polarity of the C–H bonds. Overall crystal packing of **1** and **2** is somewhat close: one can distinguish chains built from the molecules, which are arranged along Cu··· (center of $Pz^4$)···Cu mean line (Supplementary Materials Figure S4). The same type of chain was observed in the coordination polymers $[Cu(Pz^4)(NO_3)_2]_n$ [31], $[Ag(Pz^4)(NO_3)]_n$, and $\{[Ag(Pz^4)(NO_3)]DMF\}_n$ [29]. In **1** and **2**, the Cu···Cu distances within the molecule were 6.78 and 6.81 Å, while those between neighboring molecules in the chain were 7.14 and 7.94 Å, correspondingly.

### 3.3. Crystal Structure of the Ligand $Pz^4$

Conformation of the $Pz^4$ molecule in the solid state differs from that in the complexes. In free $Pz^4$, the C1–C1′–N11–N12 torsion angle (of 130.1°) characteristic for the rotation of one of the pyrazole rings around the C1′–N11 bond is obtuse (Figure 4). In contrast, all known complexes revealed acute torsion angles for both pyrazole rings due to chelate coordination of the ligand (Table 3). Note that the angle in copper(II) complexes falls within the 59°–72° range, while that in silver(I) compounds lies in the range of 50°–58°, which is in accordance with the increased ionic radius of $Ag^+$ as compared with $Cu^{2+}$.

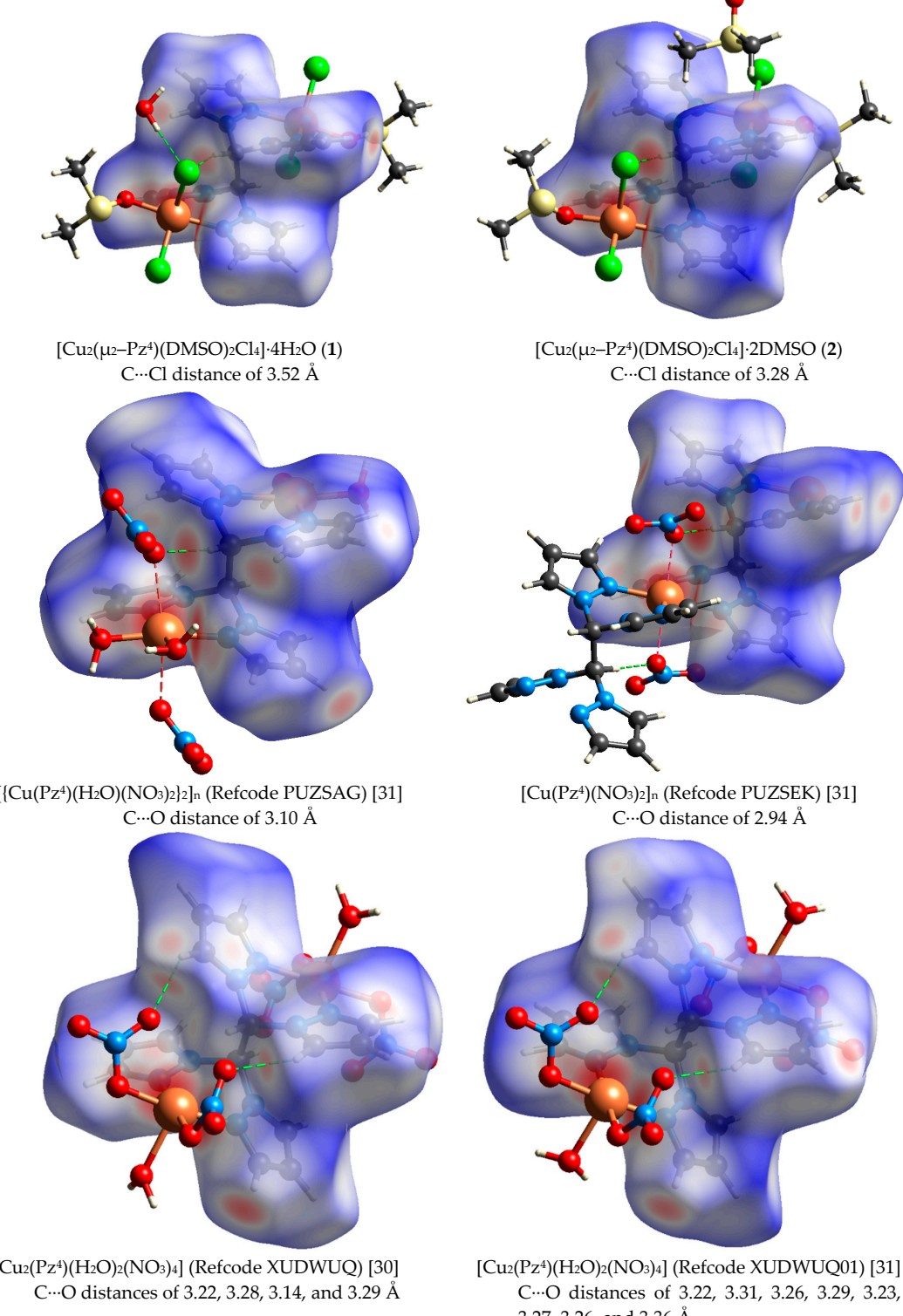

[Cu$_2$(μ$_2$–Pz$^4$)(DMSO)$_2$Cl$_4$]·4H$_2$O (**1**)
C···Cl distance of 3.52 Å

[Cu$_2$(μ$_2$–Pz$^4$)(DMSO)$_2$Cl$_4$]·2DMSO (**2**)
C···Cl distance of 3.28 Å

[{Cu(Pz$^4$)(H$_2$O)(NO$_3$)$_2$}$_2$]$_n$ (Refcode PUZSAG) [31]
C···O distance of 3.10 Å

[Cu(Pz$^4$)(NO$_3$)$_2$]$_n$ (Refcode PUZSEK) [31]
C···O distance of 2.94 Å

[Cu$_2$(Pz$^4$)(H$_2$O)$_2$(NO$_3$)$_4$] (Refcode XUDWUQ) [30]
C···O distances of 3.22, 3.28, 3.14, and 3.29 Å

[Cu$_2$(Pz$^4$)(H$_2$O)$_2$(NO$_3$)$_4$] (Refcode XUDWUQ01) [31]
C···O distances of 3.22, 3.31, 3.26, 3.29, 3.23,
3.27, 3.26, and 3.36 Å

**Figure 2.** The d$_{norm}$ Hirshfeld surface of Pz$^4$ ligand in the copper(II) complexes showing intramolecular C–H···D (D = Cl, O) contacts (marked dashed green). Area with intermolecular contacts closer than the sum of the atoms' van der Waals radii are red, longer contacts are blue, and the contacts around the sum of van der Waals radii are white. For polymeric complexes, only the ligand and coordination environment of the central atom are shown. For [Cu$_2$(Pz$^4$)(H$_2$O)$_2$(NO$_3$)$_4$] (Refcode XUDWUQ01), only one of two crystallographically independent molecules is shown.

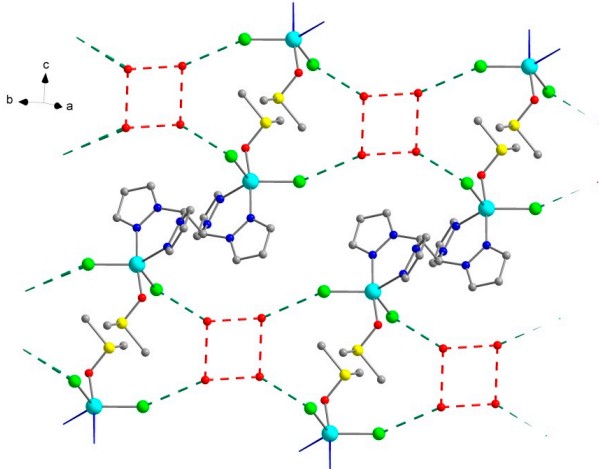

**Figure 3.** Packing diagram of [Cu$_2$(Pz$^4$)(DMSO)$_2$Cl$_4$]·4H$_2$O (**1**) showing OH···Cl (dashed green lines) and OH···O (dashed red lines) hydrogen bonds, which form a layered structure. Hydrogen atoms are omitted for clarity.

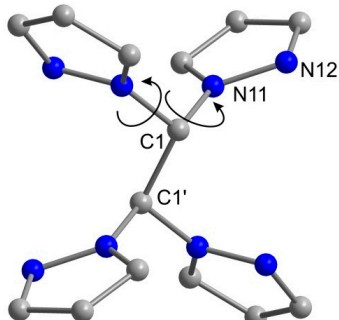

**Figure 4.** Structure of Pz$^4$, showing the C1–C1′–Nn1–Nn2 (n = 1, 2) torsion angle. Hydrogen atoms are omitted for clarity.

**Table 3.** Torsion angles of Pz$^4$ in solid state and in copper(II) and silver(I) complexes.

| Compound | C1–C1′–Nn1–Nn2 (n = 1, 2) Torsion Angle,° | References |
|---|---|---|
| Pz$^4$ | ±130.1, ±50.9 | This paper |
| [Cu$_2$(Pz$^4$)(DMSO)$_2$Cl$_4$]·4H$_2$O (**1**) | ±71,8; ±59.5 | This paper |
| [Cu$_2$(Pz$^4$)(DMSO)$_2$Cl$_4$]·2DMSO (**2**) | ±69,6; ±64.4 | This paper |
| [Cu$_2$(Pz$^4$)(H$_2$O)$_2$(NO$_3$)$_4$] | 60.8; −70.4; 69.1; −61.7 70.4; −61.5; 63.2; −67.0 | [30] |
| [Cu$_2$(Pz$^4$)(H$_2$O)$_2$(NO$_3$)$_4$] | 59.5; −74.9; 67.7; −61.0. | [31] |
| [Cu(Pz$^4$)(NO$_3$)$_2$]$_n$ | ±72.2; ±61.5; | [31] |
| [{Cu(Pz$^4$)(H$_2$O)(NO$_3$)$_2$}$_2$]$_n$ | ±70.0; ±72.5 | [31] |
| {[Ag(Pz$^4$)(NO$_3$)]DMF}$_n$ | ±57.9; ±53.8; ±50.4; ±54.8 | [29] |
| [{Ag(Pz$^4$)(NO$_3$)}$_n$] | ±57.6; ±58.1; ±50.5; ±58.2 | [29] |

## 3.4. IR Spectroscopy and DFT Calculations

For more accurate interpretation of IR spectra of the Pz$^4$ ligand and its copper(II) chloride complexes, DFT calculations of geometries of isolated molecules and normal mode vibration frequencies were carried out.

Calculated and experimental (from X-Ray single-crystal structural analysis) geometrical parameters are listed in Table 4. In most cases, deviations of calculated parameters from the experimental are rather small and probably due to crystal packing effects.

Experimental and calculated IR bands are listed in Table 5. Experimental and simulated IR spectra are shown in Figures S5 and S6. As one can see form Table 5 and Figures S5 and S6, calculated vibration frequencies are in good agreement with experimental values and can thus be used for band assignments. In the spectrum of the ligand, bands at 1520 and 1437 cm$^{-1}$ were associated with asymmetrical pyrazole

ring stretching vibrations, while the band at 1311 cm$^{-1}$ corresponded to symmetrical pyrazole stretching vibrations. Upon coordination to copper(II), one of the pyrazole stretching bands shifted to a lower frequency region (1404 and 1405 cm$^{-1}$), while two others remained at essentially the same positions. Bands of in-plane C–H vibrations (βCH) also underwent a coordination-induced low-frequency shift. It is interesting to note that the band assigned to aliphatic CCH bending vibrations demonstrated a considerable high-frequency shift from 1293 to 1471 cm$^{-1}$, which can be associated with the change of molecular symmetry that takes place upon coordination. Low-frequency bands at 254 and 260 cm$^{-1}$ in the spectra of complexes **1** and **2** were assigned to Cu–Cl stretching vibrations.

**Table 4.** Calculated and experimental geometrical parameters of Pz$^4$ and complexes.

| Parameter | Experimental | Calculated | Parameter | Experimental | Calculated |
|---|---|---|---|---|---|
| | **Pz$^4$** | | | **Complex 1** | |
| d(C7-N3), Å | 1.450(1) | 1.452 | d(Cu1-Cl1), Å | 2.436(1) | 2.426 |
| d(N3-N4), Å | 1.355(1) | 1.359 | d(Cu1-Cl2), Å | 2.314(1) | 2.323 |
| d(N4-C4), Å | 1.329(2) | 1.330 | d(Cu1-O1), Å | 1.940(3) | 2.012 |
| d(C4-C5), Å | 1.392(2) | 1.416 | d(Cu1-N1), Å | 2.074(3) | 2.212 |
| d(C5-C6), Å | 1.368(2) | 1.380 | d(Cu1-N3), Å | 2.009(3) | 2.067 |
| d(C6-N3), Å | 1.357(1) | 1.366 | d(S1=O1), Å | 1.533(2) | 1.556 |
| d(C7-C7′), Å | 1.541(1) | 1.558 | d(N3-N4), Å | 1.363(3) | 1.365 |
| | | | d(C7-C7′), Å | 1.551(3) | 1.564 |
| | | | φ(O1-Cu1-N3), ° | 171.7(1) | 168.7 |
| | | | φ(O1-Cu1-N1), ° | 83.7(1) | 83.4 |
| | | | φ(O1-Cu1-Cl2), ° | 94.09(9) | 90.3 |
| | | | φ(Cl2-Cu1-Cl1), ° | 116.67(3) | 126.0 |
| | | | φ(S1=O1-Cu1), ° | 121.0(2) | 132.5 |

**Table 5.** Calculated (scaled) and experimental vibration frequencies for Pz$^4$ and complex.

| Vibration | Pz$^4$ | | [Cu$_2$(Pz$^4$)(DMSO)$_2$Cl$_4$] | | |
|---|---|---|---|---|---|
| | $\tilde{\nu}$calc., cm$^{-1}$ | $\tilde{\nu}$exp., cm$^{-1}$ | $\tilde{\nu}$calc., cm$^{-1}$ | $\tilde{\nu}$exp., cm$^{-1}$ | $\tilde{\nu}$exp., cm$^{-1}$ |
| | | | 1 | 1 | 2 |
| νCH (Pz) | 3173 | 3137 | 3139 | 3125 | 3128 |
| νCH (Pz) | 3156 | 3130 | - | - | - |
| νCH (Pz) | 3142 | 3115 | - | - | - |
| νCH (DMSO) | - | - | 3065 | 2987 | 3000 |
| νCH | 3054 | 3018 | 2952 | 2946 | 2943 |
| νCH | 3042 | 2994 | - | - | - |
| νPz$_{asym}$ | 1509 | 1520 | 1509 | 1513 | 1513 |
| δCCH | - | - | 1458 | 1471 | 1469 |
| νPz$_{asym}$ | 1420 | 1437 | 1400 | 1404 | 1405 |
| νPz$_{sym}$ | 1292 | 1311 | 1291 | 1304 | 1302 |
| βCH (Pz) | 1380 | 1391 | 1240 | 1251 | 1254 |
| βCH (Pz) | 1203 | 1216 | 1192 | 1199 | 1200 |
| βCH (Pz) | 1154 | 1172 | 1086 | 1095 | 1095 |
| βCH (Pz) | 1073 | 1092 | 1078 | 1067 | 1068 |
| βCH (Pz) | 1029 | 1053 | 1048 | 1035 | 1032 |
| βCH (Pz) | 947 | 968 | - | - | - |
| δCCH | 1273 | 1293 | - | - | - |
| βCH (Pz) | 899 | 918 | - | - | - |
| γCH | 859 | 890 | - | - | - |
| γCH | 812 | 857 | - | - | - |
| γCH | 740 | 771 | 747 | 769 | 768 |
| γCH | 725 | 754 | - | - | - |
| δNCH | 759 | 783 | 768 | 780 | 789 |
| νSO (DMSO) | - | - | 1003 | 988 | 991 |
| νSO (DMSO) | - | - | 899 | 944 | 944 |
| γCH | 606 | 616 | 603 | 610 | 612 |
| δCCH | 569 | 585 | 548 | 550 | 549 |
| δNCC | 348 | 357 | - | - | - |
| δNCC | 302 | 319 | - | - | - |
| δNCC | 230 | 245 | - | - | - |
| δNCC | 106 | 134 | - | - | - |
| νCu-Cl | - | - | 244 | 254 | 260 |

Abbreviations: νCH (Pz)—stretching vibrations of C–H bonds in pyrazole rings; νCH (DMSO)—stretching vibrations of C–H bonds in DMSO molecules; νCH—stretching vibrations of aliphatic C–H bonds; νPz$_{asym}$, νPz$_{sym}$—asymmetrical and symmetrical pyrazole ring stretching vibrations; δCCH—bending vibrations of aliphatic C–C–H bonds; βCH (Pz)—in-plane bending vibrations of C–H bonds in pyrazole rings; γCH—out-of-plane bending vibrations of C–H bonds in pyrazole rings; δNCH—bending vibrations of aliphatic N–C–H bonds; νSO (DMSO)—stretching vibrations of S=O bonds in DMSO molecules; δNCC—bending vibrations of aliphatic N–C–C bonds; νCu-Cl—stretching vibrations of Cu–Cl bonds.

### 3.5. Thermal and XRD Analyses

The curves of thermal analysis for compounds **1** and **2** are shown in Figure 5. Investigation of thermal properties in helium atmosphere revealed that the first step of thermolysis for both compounds is associated with the removal of solvated molecules in the range of 40–150 °C for **1** and 100–200 °C for **2**. The XRD pattern for thermolysis products of **1** at 150 °C differs from the pattern of the initial compound (Figure 6), suggesting a structural rearrangement. Further decomposition leads to formation of Cu and CuCl and amorphous products of ligand degradation at 350 °C, and the final step corresponds to partial sublimation of CuCl and formation of amorphous carbon. Formation of cubic CuCl and copper phases at thermolysis temperature 350 °C was observed previously [48].

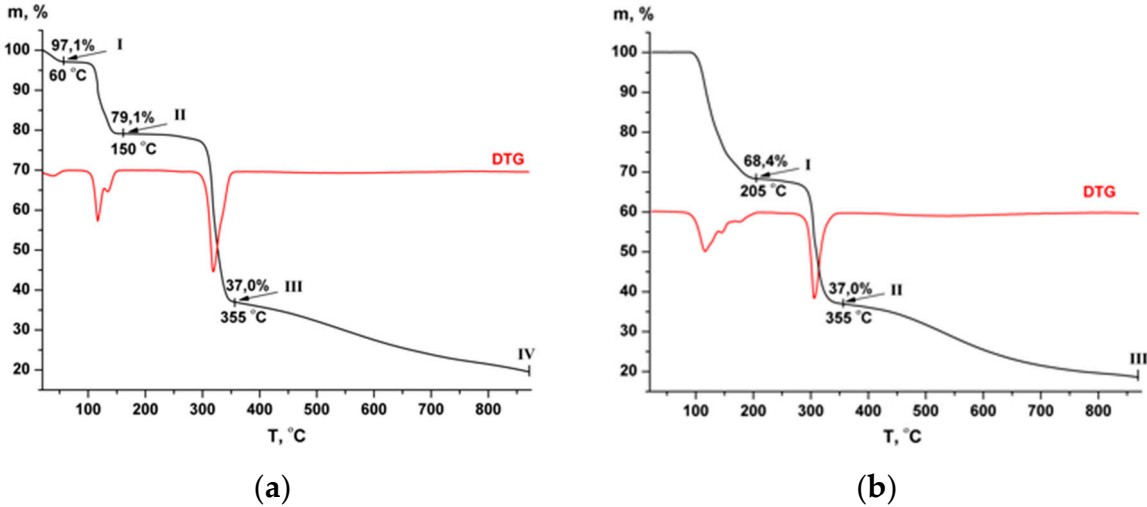

**Figure 5.** TG curves of **1** (**a**) and **2** (**b**) in inert atmosphere at 10 K/min speed rate.

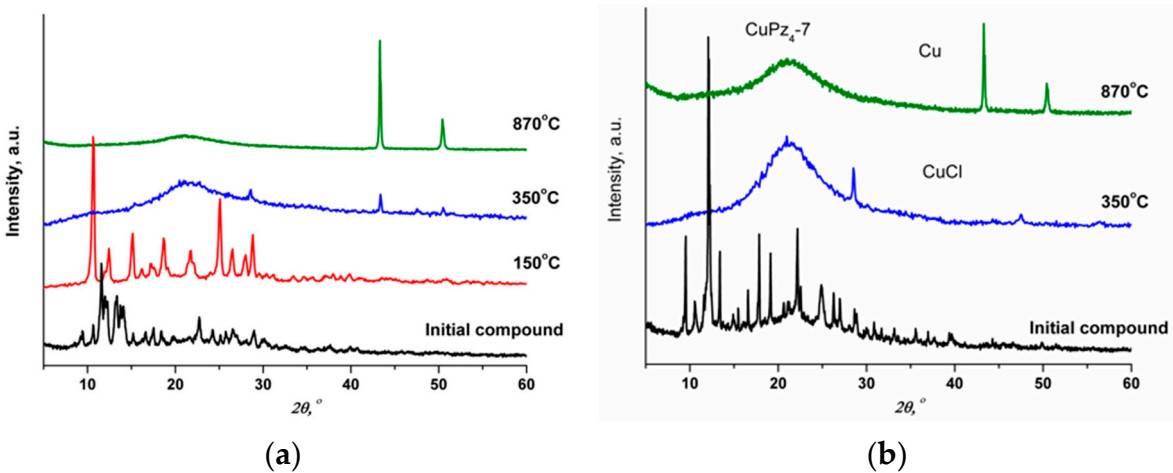

**Figure 6.** The XRD patterns for thermolysis products of complexes **1** (**a**) and **2** (**b**).

It should be noted that dehydration process of **1** proceeds in two separate steps, and it is thus possible to obtain an anhydrous compound with composition $[Cu_2(Pz^4)(DMSO)_2Cl_4]$. Thermal decomposition of **2** runs in a more complex manner with a simultaneous loss of outer- and inner-sphere dimethyl sulfoxide molecules in one step (Figure 5). Concurrent loss of both outer- and inner-sphere DMSO molecules can be explained by the absence of intermolecular interactions between solvent molecules in complex **2** in contrast to the hydrogen-bonded network in complex **1**.

## 4. Conclusions

Two binuclear coordination compounds of Cu(II) chloride with 1,1,2,2-tetrakis(pyrazol-1-yl)ethane (Pz$^4$) of the composition [Cu$_2$($\mu_2$-Pz$^4$)(DMSO)$_2$Cl$_4$]·4H$_2$O and [Cu$_2$($\mu_2$-Pz$^4$)(DMSO)$_2$Cl$_4$]·2DMSO have been synthesized and fully characterized by elemental analysis, IR spectroscopy, thermogravimetric analysis, single-crystal X-ray diffraction, and powder diffraction analysis. Crystal structure analysis of the compound [Cu$_2$($\mu_2$-Pz$^4$)(DMSO)$_2$Cl$_4$]·4H$_2$O revealed a layered supramolecular structure formed by OH···Cl and OH···O (dashed red lines) hydrogen bonds.

**Supplementary Materials:** The following are available online at http://www.mdpi.com/2073-4352/9/4/222/s1, Table S1: Atomic coordinates for optimized geometry of complex **1**, Table S2: Atomic coordinates for optimized geometry of ligand Pz$^4$, Figure S1: Overlay of the metallocycles of **1** and **2**, [Cu$_2$(Pz$^4$)(H$_2$O)$_2$(NO$_3$)$_4$] (refcode XUDWUQ), [{Cu(Pz$^4$)(H$_2$O)(NO$_3$)$_2$}$_2$]$_n$ (refcode PUZSAG), [Cu(Pz$^4$)(NO$_3$)$_2$]$_n$ (refcode PUZSEK) and [Cu$_2$(Pz$^4$)(H$_2$O)$_2$(NO$_3$)$_4$] (refcode XUDWUQ01), Figure S2: Overlay of the molecules **1** and **2,** Figure S3: The d$_{norm}$ Hirshfeld surface of the molecules **1** and **2** showing intermolecular C–H···D and O–H···D (D = Cl, O) contacts, Figure S4: Crystal packing of **1** and **2** with a view along the chains built from the molecules, Figure S5: Calculated and experimental IR spectra of the ligand Pz$^4$ in the characteristic 1600–500 cm$^{-1}$ range, Figure S6: Calculated and experimental IR spectra of compound **1** in the characteristic 1600–500 cm$^{-1}$ range. CCDC 1900906–1900908 contain the supplementary crystallographic data for this paper. These data can be obtained free of charge from the Cambridge Crystallographic Data Centre via www.ccdc.cam.ac.uk/data_request/cif.

**Author Contributions:** Conceptualization, A.P.; investigation, E.L., T.S., A.S., E.S. and E.F.; writing—original draft, E.L., T.S., and E.S.; writing—review and editing, A.P.

**Funding:** The reported study was supported by the Russian Science Foundation (Project No. 18-73-00294).

**Acknowledgments:** DTF calculations were carried within the framework of the Tomsk Polytechnic University Competitiveness Enhancement Program. The authors are thankful to Dr. Enrico Benassi for assistance with DFT calculations.

**Conflicts of Interest:** The authors declare no conflict of interests.

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
