# Peer review of "Synthesis, Crystal Structure, Thermal Analysis, and DFT Calculations of Molecular Copper(II) Chloride Complexes with Bitopic Ligand 1,1,2,2-tetrakis(pyrazol-1-yl)ethane"

_crystals, doi:10.3390/cryst9040222_

Round 1

Reviewer 1 Report

The report concerns the crystallographic study of two binuclear coordination compounds. It is fairly written and easy to read. However, I can't find any noble point in the manuscript. Similar structures with copper ions have been already reported. I can't believe this result is noble enough to consider its publication. Also, structural description is too short and should be improved with comparison with previously reported structures.

Author Response

1. The report concerns the crystallographic study of two binuclear coordination compounds. It is fairly written and easy to read. However, I can't find any noble point in the manuscript. Similar structures with copper ions have been already reported. I can't believe this result is noble enough to consider its publication. Also, structural description is too short and should be improved with comparison with previously reported structures.

We would like to point out, that the only previously reported structure of copper complexes of Pz4 ligand is copper(II) nitrate complexes (reported by our group, reference 30 in the manuscript). We have extended the structural part of the manuscript, which now includes the use of geometrical criteria to assign the types of coordination arrangements of copper ions in both of the complexes. These criteria actually suggest that coordination polyhedra in complexes 1 and 2 are actually different and we now point this out in section 3.2 as well as compare the structures of the coordination spheres with a previously reported copper nitrate complex.

Reviewer 2 Report

The authors describe syntheses, structures, and some properties of two binuclear copper complexes. I found the work to be competently done, and the paper is well written. Therefore, I suggest this manuscript to be accepted after the following issues have been adequately addressed.

I do not understand the reason why complex 1 and 2 are prepared separately because both complexes have the same composition in the coordination sphere. The authors should clearly explain the reason in the text (section 3.1).

Other minor points:

Introduction (line 36): The authors must define the ligand (Pz4) in this part, not line 45.

Introduction (line 37): … activity of Pz4 ligand [21]. → … activity of Pz4 [21].

Materials and Methods (line 64): The number of complexes (1 and 2) should be defined in the Introduction section.

Figure 1 (line 137): Solvent molecules and hydrogen atoms are omitted for clarity.

3.4 IR and DFT calc. (line 169): In is interesting … → It is interesting …

Author Response

1. I do not understand the reason why complex 1 and 2 are prepared separately because both complexes have the same composition in the coordination sphere. The authors should clearly explain the reason in the text (section 3.1).

Although the complexes 1 and 2 differ only in outer-sphere solvate molecules, they demonstrated different crystal packing and different thermal behavior. This explanatory note was added in the end of section 3.1.

Other minor points:

2. Introduction (line 36): The authors must define the ligand (Pz4) in this part, not line 45.

Corrected

3. Introduction (line 37): … activity of Pz4 ligand [21]. → … activity of Pz4 [21].

Corrected

4. Materials and Methods (line 64): The number of complexes (1 and 2) should be defined in the Introduction section.

Complexes 1 and 2 have the same inner sphere, but different solvate molecules, which lead to different supramolecular structure, therefore we describe both of them separately.

5. Figure 1 (line 137): Solvent molecules and hydrogen atoms are omitted for clarity.

corrected

6. 3.4 IR and DFT calc. (line 169): In is interesting … → It is interesting …

Corrected

Reviewer 3 Report

This paper is considered to be acceptable because the crystal structures of the ligand 1,1,2,2-tetrakis(pyrazol-1-yl)ethane (Pz4) and its copper(II) complexes were newly determined and thermal and XRD analyses as well as IR measurements with DFT analyses were also performed, although it is pity that the magnetic and UV-vis spectral data in solid are not given.

Some comments are itemized below:

At 36 line: The definition for Pz4 should be given.

At 103 line:  “vile” should be “vial”.

At 113 line:  Scheme 1 is not effective, because the difference between 1 and 2 is the crystal solvent (DMSO or H2O).  The Cu(II) dinuclear core structure should be only described.

At 124 line: “trans-positions” should be “axial positions”.

In Table 1, two values are given for the torsion angle (in the case of Pz4; 1030.1, 50.1).  Both the values should be discussed in the text.

At 169 line: “In is interesting to…” should be “It is interesting…”.

At 193 line: Why is the decomposition process of 2 complicated compared with that of 1.?  Why does the simultaneous loss occur?  The authors should explain the reasons using the crystal data.

Author Response

1. At 36 line: The definition for Pz4 should be given.

Corrected

2. At 103 line:  “vile” should be “vial”.

Corrected

3. At 113 line:  Scheme 1 is not effective, because the difference between 1 and 2 is the crystal solvent (DMSO or H2O).  The Cu(II) dinuclear core structure should be only described.

Scheme 1 was reformatted to show both solvates in one formula.

4. At 124 line: “trans-positions” should be “axial positions”.

Corrected

5. In Table 1, two values are given for the torsion angle (in the case of Pz4; 1030.1, 50.1).  Both the values should be discussed in the text.

In section 3.3 we indicate that upon coordination rotation of one of pyrazole rings takes place and torsion angle changes from 130 to 60-70 degrees. The second pyrazole ring in the crystal structure of the free ligand is already oriented in a way ready for chelation and thus its torsion angle does not change much. The discussion text in section 3.3 was modified to make the difference between two angles clear.

6. At 169 line: “In is interesting to…” should be “It is interesting…”.

corrected

7. At 193 line: Why is the decomposition process of 2 complicated compared with that of 1.?  Why does the simultaneous loss occur?  The authors should explain the reasons using the crystal data.

Concurrent loss of both outer and inner sphere DMSO molecules can be explained by the absence of intermolecular interactions between solvent molecules in complex 2 in contrast to hydrogen-bonded network in complex 1. A note was added to the manuscript on lines 196-198.

Round 2

Reviewer 1 Report

I still can't accept its publication in journal Crystals. Structural description is too short and there is no comparison with previously reported copper complexes.(I am not talking about the simple comparison of tau value of copper metal centers) Detailed comparison of connectivity of previously reported copper complexes, 1 and 2 with visualization should be included in the main text.

Author Response

We have substantially extended the structural part of the manuscript, which now includes description of intra- and intermolecular interactions using Hirshfeld surface analysis, as well as comparison of crystal structure features with our previously reported compounds, the additions are highlighted in green.

Round 3

Reviewer 1 Report

Now I suggest its publication in Crystals.